# Incorporating Robustness to Imaging Physics into Radiomic Feature Selection for Breast Cancer Risk Estimation

**DOI:** 10.3390/cancers13215497

**Published:** 2021-11-01

**Authors:** Raymond J. Acciavatti, Eric A. Cohen, Omid Haji Maghsoudi, Aimilia Gastounioti, Lauren Pantalone, Meng-Kang Hsieh, Emily F. Conant, Christopher G. Scott, Stacey J. Winham, Karla Kerlikowske, Celine Vachon, Andrew D. A. Maidment, Despina Kontos

**Affiliations:** 1Department of Radiology, Perelman School of Medicine, University of Pennsylvania, Philadelphia, PA 19104, USA; Eric.Cohen@pennmedicine.upenn.edu (E.A.C.); o.maghsoudi@gmail.com (O.H.M.); a.gastounioti@wustl.edu (A.G.); Lauren.Pantalone@pennmedicine.upenn.edu (L.P.); michaelhsieh42@gmail.com (M.-K.H.); Emily.Conant@pennmedicine.upenn.edu (E.F.C.); Andrew.Maidment@pennmedicine.upenn.edu (A.D.A.M.); Despina.Kontos@pennmedicine.upenn.edu (D.K.); 2Division of Epidemiology, Department of Health Sciences Research, Mayo Clinic, Rochester, MN 55905, USA; Scott.Christopher@mayo.edu (C.G.S.); Winham.Stacey@mayo.edu (S.J.W.); Vachon.Celine@mayo.edu (C.V.); 3Departments of Medicine and Epidemiology/Biostatistics, Women’s Health Clinical Research Center, UCSF, San Francisco, CA 94143, USA; karla.kerlikowske@ucsf.edu

**Keywords:** radiomics, digital mammography, robustness, feature selection, anthropomorphic phantom, case-control analysis, risk assessment, imaging acquisition physics, breast cancer

## Abstract

**Simple Summary:**

Mammographic density estimates can be combined with radiomic texture features to offer an even better assessment of breast cancer risk. However, some feature variations will be due to true parenchymal differences between women, but others will be due to imaging physics effects (contrast, noise, and image sharpness); features robust to imaging physics effects should better model risk. To investigate this, we imaged an anthropomorphic phantom at various x-ray technique settings, allowing us to directly measure the effects of imaging physics on feature values. We compared these variations, for each feature, with the inter-woman variation in a screening population (552 cancer-free women) and the intra-woman variation between each woman’s left and right breasts, to assess which features were relatively robust to physics settings. We then tested more- versus less-robust features in modeling cancer risk on an independent case-control data set, and demonstrated that more-robust features were indeed better at risk prediction.

**Abstract:**

Digital mammography has seen an explosion in the number of radiomic features used for risk-assessment modeling. However, having more features is not necessarily beneficial, as some features may be overly sensitive to imaging physics (contrast, noise, and image sharpness). To measure the effects of imaging physics, we analyzed the feature variation across imaging acquisition settings (kV, mAs) using an anthropomorphic phantom. We also analyzed the intra-woman variation (IWV), a measure of how much a feature varies between breasts with similar parenchymal patterns—a woman’s left and right breasts. From 341 features, we identified “robust” features that minimized the effects of imaging physics and IWV. We also investigated whether robust features offered better case-control classification in an independent data set of 575 images, all with an overall BI-RADS^®^ assessment of 1 (negative) or 2 (benign); 115 images (cases) were of women who developed cancer at least one year after that screening image, matched to 460 controls. We modeled cancer occurrence via logistic regression, using cross-validated area under the receiver-operating-characteristic curve (AUC) to measure model performance. Models using features from the most-robust quartile of features yielded an AUC = 0.59, versus 0.54 for the least-robust, with *p* < 0.005 for the difference among the quartiles.

## 1. Introduction

Breast density has consistently been shown to be an independent predictor of breast cancer risk [1,2,3]. Recent studies have demonstrated that combining estimates of breast density with radiomic features obtained from digital mammography (DM) images results in an improved discrimination of breast cancer risk, and that these calculations have important applications in developing personalized screening and prevention strategies for breast cancer [4,5,6].

The number of potentially predictive radiomic features is constantly growing. However, risk-assessment models are not necessarily improved with the use of more features, since features can be highly-correlated with each other. Previous works have proposed dimensionality reduction techniques to filter out correlated features based on cluster analysis and principal component analysis [7,8,9,10,11,12,13,14].

As an independent (but complementary) filtering strategy, one can identify the features that are robust to variations in imaging physics. The goal is to identify features that are not overly sensitive to contrast, noise, and image sharpness, which may change due to variations in imaging acquisition settings, scanner make and model, image processing, or even software version. Instead, it is important to select features that capture underlying parenchymal and pathophysiological differences between women, which could be suggestive of breast cancer risk.

Previous works have analyzed women imaged under different conditions (for example, with different DM vendors) to identify robust features. Mendel et al. found that features measuring spatial patterns were insensitive to vendor (for example, fractal dimension, power law β, and correlation from the co-occurrence matrix) [15]. However, intensity- and directionality-based features were sensitive to vendor (for example, entropy, balance, and mean gradient). In a follow-up study, this analysis was applied in combination with hierarchical clustering to identify features that were robust across vendors and also non-redundant [14].

Although it is ideal to identify robust features based on data sets of women imaged under different conditions, these data sets are rare and become quickly obsolete with the changing imaging technologies. An alternative is to use an anthropomorphic phantom which has the advantage of being able to be imaged under an unlimited number of conditions, so that the effects of imaging physics can be measured directly. Keller et al. showed that a phantom (Gammex 169, “Rachel”, Madison, WI, USA) can successfully be used to measure the robustness of features across different DM vendors [16,17]. In their work, each image was Z-score normalized, and feature distributions across vendors were analyzed with equivalence tests to identify robust features.

We extended this prior work to investigate the effect of varying the x-ray technique settings (kV and mAs) using the same phantom as in Keller’s study, but with a larger number of features (341 features as opposed to 29). Additionally, unlike all the works described previously, we identified robust features using acquisitions from a single vendor. For each feature, a robustness metric called imaging acquisition variation (IAV) was calculated; it measures how much the variation due to x-ray technique settings (kV and mAs) in a phantom scales against the variation across women in a screening population. All images in this screening population had an overall assessment of 1 (negative) or 2 (benign) according to the Breast Imaging Reporting and Database System (BI-RADS^®^). The women were validated to be cancer-free with at least 12 months of follow-up in a subsequent screening exam with an overall BI-RADS^®^ assessment of 1 or 2. This clinical data set was derived from only one vendor; our method for identifying robust features does not require women to have been imaged with different technique settings or different DM vendors, as was the case in the work of Mendel et al. [15]. In order for a feature to be considered robust, the variation across technique settings in a single phantom should be small compared against the variation across a population of women, where there are broadly differing parenchymal patterns.

In addition, we calculated intra-woman variation (IWV) for each feature using the same data set of clinical mammograms. IWV is a measure of how much the feature varies across the left and right breasts of the same woman. For a feature to provide clinically-meaningful information about breast cancer risk, it is assumed that the feature should not be highly sensitive to left–right breast differences in the same woman, but instead, be more sensitive to differences across a population of women. We identified features that are overly sensitive to bilateral differences in the same woman so that these features are excluded in risk-assessment analyses, just as features that are highly sensitive to the effects of imaging physics can be excluded. For this purpose, we developed a composite measure of variation (CMV) that combines both IAV and IWV, with the goal of using CMV to identify robust features.

Because CMV is calculated exclusively with mammograms from women who were validated to be cancer-free (overall BI-RADS^®^ assessment of 1 or 2) in a subsequent screening exam, it was necessary to consider a separate clinical data set to investigate the relationship between CMV and case-control classification performance. For these calculations, there were four classes of features (quartiles), ranked in order of robustness scores (CMV). We built a logistic regression model for case-control classification with nine randomly-selected features in each class using cross-validated area under the receiver-operating-characteristic curve (AUC) to measure model performance. Since a relatively small number of features (nine) was utilized in this AUC calculation and since established risk factors for breast cancer (e.g., age, breast density, and body mass index) were not modeled, it should be emphasized that the purpose of these calculations was not to develop a model with the highest possible AUC. Rather, the aim was to demonstrate a statistically-significant improvement in AUC in models built with more robust features. Ultimately, we did indeed demonstrate that there is a trend of better discrimination in models built using more robust features, and therefore, we conclude that a robustness analysis is an essential step in selecting features for risk-assessment modeling.

## 2. Materials and Methods

### 2.1. Roadmap

This study consisted of two components. First, we developed a metric (CMV) which measures the robustness of each feature (Section 2.2). The metric captures two sources of undesirable feature variation: the variation due to differences in imaging physics settings; and intra-woman (left–right) variation that is unlikely to be related to meaningful breast differences. In developing this metric, we employed a dataset of women with negative or benign screening exams, for which there was a subsequent negative or benign follow-up exam at least one year later. Second, we validated the utility of the CMV metric in a case-control classification, testing whether more-robust features give superior classification performance (Section 2.3). This analysis employed an independent case-control data set. In this, all mammograms are negative or benign, but the population includes cases—women who developed cancer at least one year later.

### 2.2. Robust Feature Identification

#### 2.2.1. Study Population

This study was HIPAA-compliant and approved by the University of Pennsylvania Institutional Review Board with a waiver of consent under expedited review category 5 (Review Board IRB #7, Protocol #825735). We retrospectively analyzed DM screening images from 997 women acquired at one facility, sequentially from 2 September through 31 October 2014; this represents a subset of a larger cohort described in a previous work [18]. All images had an overall BI-RADS^®^ assessment of 1 or 2; each woman had a similar overall BI-RADS^®^ assessment in a subsequent screening exam at least 12 months later. The women were imaged with Selenia Dimensions systems (Hologic Inc., Bedford, MA, USA). We analyzed only cranial–caudal (CC) view images of breasts with thickness in [40, 60] mm to compare to breast properties to those of a phantom modeling a 50-mm thick breast in CC view.

#### 2.2.2. Phantom Data Acquisitions

DM images of the phantom were acquired in a CC projection using a Selenia Dimensions system in “Manual” mode over a range of kV and mAs combinations (examples shown in Appendix A). Two target/filter combinations are supported by this system: W/Rh and W/Ag. According to Hologic data tables for automatic exposure control settings, breasts with thickness in [40, 60] mm are imaged with W/Rh, thus all phantom images were acquired with this target/filter combination. First, images were acquired between 27 and 35 kV in 1 kV increments, with mAs determined with auto timing (Appendix A). Second, at the central kV (31), we varied the mAs between 13 and 180 mAs in increments varying by a factor of 2^1/2^ or 1.41 (at the closest mAs supported by the system). The phantom was imaged twice at each technique setting.

The Appendix A illustrate the trends for two features across the full range of kV and mAs settings (Appendix A), but our goal was to develop a robustness metric that captured the effect of kV and mAs variations that might be seen clinically. To do this, we identified the (kV, mAs) settings represented in our clinical image dataset (images of breasts similar in thickness to the phantom), and kept only the phantom images acquired at settings in these bounds (Appendix A). Six technique settings met this criterion: 28 kV (160 mAs), 29 kV (140 mAs), 30 kV (120 mAs), and 31 kV (120, 140, and 180 mAs). Since each acquisition was repeated, a total of 12 images were analyzed.

#### 2.2.3. Feature Extraction

Texture features were calculated on raw (“FOR PROCESSING”) images; automatic segmentation of the breast area was performed using the publically available LIBRA software package (version 1.0.4) [19]. Partitioning the breast into a lattice of 6.3 mm square windows, for a given feature, the feature value was calculated within every window, and then averaged over all the windows. This approach was motivated by previous work by Zheng et al., which showed that the use of a lattice results in better case-control classification than the use of a single region of interest [4]. Zheng et al. also studied the effect of varying the window size (between 6.3 and 25.5 mm), and found that the area under the receiver operating characteristic curve (AUC) was highest at the 6.3-mm window size.

A total of 341 features were analyzed: 12 grey-level histogram, 7 co-occurrence, 7 run length, 2 fractal dimension, 36 local binary pattern (LBP), 125 Laws, 120 co-occurrence Laws, and 32 Gabor wavelet features [20,21,22,23,24].

#### 2.2.4. Imaging Acquisition Variation

To measure how much each feature was affected by imaging physics, the range of variation across 12 phantom acquisitions was calculated, and scaled against the spread of the clinical distribution (Figure 1). In a robust feature (Figure 1a), the phantom values are clustered over a narrow range of values relative to the variation in the population. By contrast, in a feature that is strongly affected by imaging physics (Figure 1b), the phantom data points are spread across a broad range.

This motivated our first feature quality metric, imaging acquisition variation (IAV). The range of phantom variation is scaled against the width of the middle 90% of the clinical distribution. (This width was chosen, as opposed to the full range of the distribution, as the tails could be sensitive to outliers.) Denoting *p_ik_* as the value of the *k*th feature for the *i*th phantom acquisition, IAV was thus defined as:(1)$${\mathrm{IAV}}_{k}=\frac{\max_{i}\{p_{ik}{\;\}}_{i=1}^{m}-\min_{i}\{p_{ik}{\;\}}_{i=1}^{m}}{c_{95,k}-c_{5,k}}$$
where *i* runs from 1 to the number of phantom acquisitions (*m* = 12), *k* runs from 1 to the number of features (341), and *c*_95,*k*_ and *c*_5,*k*_ (dashed vertical lines, Figure 1) denote the 95th and 5th percentile of the clinical distribution for the *k*th feature. In a robust feature, IAV should effectively be zero, since variations due to imaging physics (the numerator) should represent a much smaller effect than variations between women (the denominator).

#### 2.2.5. Intra-Woman Variation

A second feature quality metric, the intra-woman variation (IWV), was calculated based on an analysis of bilateral (left–right) differences in each woman. Just as IAV should effectively be zero, one would expect IWV to be minimized in a robust feature, since parenchymal patterns are generally similar across a woman’s two breasts. Denoting *l_jk_* as the value of the *k*th feature in the left breast of the *j*th woman and *r**_jk_* as the corresponding value in her right breast, IWV was then defined as:(2)$${\mathrm{IWV}}_{k}=\frac{{\mathrm{median}}_{j}{\{|l}_{jk}-r_{jk}{\left|\right\}}_{j=1}^{q}}{c_{95,k}-c_{5,k}}$$
where *q* is the number of women with at least one breast with thickness in [40, 60] mm in CC view. (We chose the median as the summary measure over all women as less sensitive to outliers than the mean.) The denominator is the same as that for IAV, scaling the value to the feature’s variation over all breasts with thickness in [40, 60] mm in CC view.

#### 2.2.6. Composite Measure of Variation

The two feature quality metrics were visualized for all 341 features with a scatter plot in IAV × IWV space. More robust features are closer to the origin (the point for which IAV = IWV = 0) in this space. For this reason, we calculated the Euclidean distance of each data point from the origin as a composite measure of variation (CMV), a metric that takes into account the combined influence of IAV and IWV:(3)$${\mathrm{CMV}}_{k}=\sqrt{{\mathrm{IAV}}_{k}{}^{2}+{\mathrm{IWV}}_{k}{}^{2}}$$
where 0 ≤ CMV*_k_* < ∞. The purpose of CMV was to capture, in a single metric, how much each feature is sensitive to imaging physics as well as left–right differences in the same woman, with the ultimate goal of using CMV to identify features that minimize both sources of variation.

### 2.3. Case-Control Analysis

If a feature is less sensitive to changes in imaging settings, this may make it less sensitive to the variations in facility, device manufacturer, and protocol that are found in real-world imaging data; and this feature may then offer more information and less noise when used in modeling on a heterogeneous data set. We therefore also analyzed whether modeling cancer risk using lower-CMV features tended to perform better than the same modeling using higher-CMV features.

#### 2.3.1. Study Population

For this purpose, we analyzed an independent case-control data set described by Gastounioti et al. [25]. It consisted of 575 images with an overall BI-RADS^®^ assessment of 1 or 2, with a subset of 115 images (cases) known to have developed cancer at least one year later. As described in Gastounioti’s work, the remaining 460 images were controls matched on the basis of age, ethnicity, and screening exam date (within one year), who were known to have an overall BI-RADS^®^ assessment of 1 or 2 in a follow-up screening exam at least 12 months later. We analyzed the raw (“FOR PROCESSING”) DM images, which were acquired with Selenia Dimensions systems. All images were mediolateral-oblique (MLO) views, with the exception of two CC views.

#### 2.3.2. Dimensionality Reduction

First, recalling the data set of CC views of 997 women (the data set used to calculate robustness metrics IAV, IWV, and CMV), we analyzed the clinical distribution of each feature as the basis for an initial dimensionality reduction step. We Z-score normalized each distribution, and excluded features with highly-skewed (skewness >6) and highly asymmetric (kurtosis >50) distributions. All CC views independent of thickness were used in this analysis.

To remove highly-correlated features, we then identified pairs of features with >95% Pearson’s correlation. For each pair, the feature with the lower inter quartile range (IQR) was excluded (similar to the approach used by Gastounioti et al.) [25].

#### 2.3.3. Case-Control Classification

Next, the remaining features were ranked by CMV and partitioned, by quartiles, into four feature robustness classes, A (lowest CMV) through D (highest). Our aim then was to test whether models built on features from class A would, on average, predict case-control status better than models built on features from class D (with the other two classes falling in between). We divided the set of features into quartiles, rather than contrasting robust with nonrobust features via a numeric cutoff on CMV, as quartiles allowed a consistent evaluation, with the same number of features in each comparison group. Using more than two feature groups allowed for better discovery of any possible trend in performance versus robustness, and four quartiles permitted an adequate number of features in each group.

To do this, we employed the features in logistic regression models, with five-fold cross-validated AUC as the measure of model performance: for each model, four-folds (460 women—92 cases, 368 controls) were employed for modeling, and the remaining fold (115 women—23 cases, 92 controls) for calculating AUC. (Case-control matching was preserved in all partitions.) Since early investigation showed that the partitioning used for cross-validation had a substantial impact on AUC, we (a) tested each model on ten different random partitions of the data, to smooth this effect over multiple partitions; and (b) used the same ten partitions for all models, to provide an even playing field across models and feature classes. In all, we built ten models for each feature class, each model using nine features randomly selected from that class; tested each model ten times, each time using a different one of the ten data partitions; and recorded these 100 five-fold cross-validated AUCs as performance measures for each feature class (Figure 2). (Each logistic regression model employed nine features as, with 115 cases, and four of the five folds used to build each model, this was the maximum possible allowing a rule-of-thumb 10 cases for each coefficient in the model).

Finally, those 400 values were used to build a linear mixed model testing against the null hypothesis that AUC was unrelated to class, with mean cross-validated AUC as the outcome, treating robustness class as a fixed effect (with reference level D) and partition as a random effect. Selecting a reference class permitted linear regression versus mean AUC, and employing class D as the reference allowed us to understand the gain in AUC versus the performance of the least-robust features. This analysis was done at the 0.05 level of significance. All statistical analysis was performed using the R language version 4.0.0 and Python version 3.6.0 with the following installed packages: Numpy 1.18.5, Scipy 1.4.1., Pandas 1.0.3, statsmodels 0.11.1, matplotlib 3.3.0, and sklearn 0.23.0 [26].

## 3. Results

### 3.1. Roadmap

Our two research aims yield two key results. First, we developed the CMV metric, deriving it from two underlying metrics, IAV and IWV, measuring imaging physics effects and intra-woman variation (Section 3.2). This allowed us to rank the features in their overall robustness. Second, we validated the utility of the CMV metric in a case-control classification on an independent data set (Section 3.3), finding a trend of better modeling performance in features with lower CMV.

### 3.2. Robust Feature Identification

#### 3.2.1. Study Population

The data set of CC views of 997 women yielded 1984 raw CC mammograms that successfully underwent automated segmentation and texture feature extraction. Restricting these to only images of breasts in [40, 60] mm thickness resulted in 970 images representing 552 women (Table 1). Of these, 968 images were acquired with W/Rh target/filter combination and two images (from the same patient) were acquired with W/Ag target/filter combination. Images derived from both target/filter combinations were used in all calculations and analyses.

#### 3.2.2. Robustness Calculations

Most IAV scores were clustered around zero, with a long tail corresponding to features highly sensitive to imaging physics: range = 0–1.85, median = 0.12, mean = 0.20, IQR = 0.04–0.19 (Figure 3a). IWV scores, however, showed a peak above zero, indicating that some difference in texture between the two lateralities of the same woman is normal. IWV range = 0–0.4, median = 0.09, mean = 0.10, IQR = 0.08–0.11 (Figure 3b).

In the IAV × IWV space (Figure 4), each feature corresponds to a data point from which we calculated CMV; i.e., Euclidean distance from the origin (the point for which IAV = IWV = 0). More robust features have lower CMV scores, approaching zero. In the Appendix A, the distribution of CMV scores is analyzed explicitly in each of the eight feature families (Appendix A).

For this work, a strict cut-off dichotomizing features as either robust or nonrobust is not being proposed. However, if one were to accept 15% non-meaningful variation as the maximum for either clinical use or research, the circular arc in Figure 4 shows a CMV threshold of 0.15 as an example that might be chosen. There are 158 features of 341 (46%) that meet this threshold. Depending on the application, a more conservative or more liberal threshold might be chosen.

#### 3.2.3. Examples of Robust and Non-Robust Features

Heatmaps of feature intensity within a breast can show the effects of imaging acquisition physics. Heatmaps of a feature in clinical images, corresponding to the 5th, 50th, and 95th percentiles of the histograms (Figure 1), show clear differences between those breasts (Figure 5a–c and Figure 6a–c). Heatmaps of the phantom, at the extremes and middle value of physics settings (Figure 1), show almost no variation in a robust feature (CMV = 0.11) (Figure 5d–f). However, in a feature that is strongly affected by imaging physics (CMV = 1.43), differences are evident among heatmaps of that feature in the phantom under different imaging settings (Figure 6d–f).

### 3.3. Case-Control Evaluation

#### 3.3.1. Dimensionality Reduction

After eliminating the highly-skewed, highly-tailed, and highly-correlated features, we retained 109 features. It was necessary to exclude one feature to ensure equal sized feature robustness quartiles, thus we excluded the highest-CMV of the 109. In the Appendix A, we summarize (by feature family) which of the 341 features were retained (Appendix A).

#### 3.3.2. Performance Evaluation

In the independent case-control data set, there were 115 cases, including 86 invasive cancers (75%) and 29 in situ cancers (25%). The remaining 460 images were matched controls (Table 2). There was an overall trend to higher AUC (better case-control classification) in models built using more robust features, with mean (SD) AUC for classes D–A of 0.54 (0.029), 0.56 (0.023), 0.56 (0.029), and 0.59 (0.023) (Figure 7).

The linear mixed model regression of AUC on feature class, with class D (the least robust features) as the reference class, gave *p* < 0.005 for a difference between each of classes A through C and the reference class (Table 3).

## 4. Discussion

As the number of features used in radiomics studies continues to grow, it is essential to understand whether features capture information about parenchymal and pathophysiological differences between women, and/or measure variations in imaging acquisition physics (contrast, noise, and image sharpness). In this study, we investigated how much a feature varies across imaging acquisition settings (kV and mAs) in DM. It is not practical or ethical to image the same woman multiple times under various acquisition settings. An alternate approach is to perform these experiments with a phantom. We analyzed a phantom that was derived from an actual woman’s mammogram [16]. The x-ray experiments in this study were equivalent to imaging that woman under various technique settings. These acquisitions included three different physics effects: (1) variation in kV (contrast); (2) variation in mAs (relative signal to noise); and (3) image replication (quantum noise).

Although a phantom is essential for understanding the effect of imaging acquisition physics, the phantom alone is insufficient for identifying robust features. Combining the phantom results with a large clinical data set allowed us to measure how much the variation across technique setting scales against the variation across women. In addition, the clinical data set allowed us to calculate IWV for each feature, so that we could identify features that are insensitive to both imaging physics effects and to left–right differences in the same woman.

Our previous work showed that radiomic features may vary with the compressed breast thickness [27]. This observation motivated us to define the robustness metrics (Equations (1)–(3)) in such a way as to capture information from only a subset of women (thickness of ±10 mm relative to the phantom in CC view). The purpose of restricting the thickness range was to ensure that the denominator of IAV (Equation (1)) is not skewed in some features that may be more strongly sensitive to thickness (given that the numerator models only one thickness, 50 mm, in the phantom). We are cognizant that this approach is a potential limitation of this study and that future work also needs to explore how the metrics IAV and IWV would change if a different phantom or a different clinical data set were chosen for the analysis (differing, for example, in terms of ethnicity, age, breast density, and body mass index).

Prior works have proposed the idea of standardizing feature calculations across different acquisition settings. For example, Andrearczyk et al. used neural networks and domain adversarial training to standardize radiomic features across computed tomography scanners with different protocols using a texture phantom [28]. Our current study offers a different perspective on understanding a feature’s variation across imaging acquisition settings. The emphasis of our current study is not on developing a transformation to standardize the feature’s values across different acquisition settings, but instead, on identifying which features are least sensitive to these settings, so that risk-assessment models in future studies could be built exclusively with robust features.

With an independent case-control data set, we demonstrated a statistically-significant increase in AUC in logistic regression models built with lower-CMV (more robust) features. Based on these results, we project that models built exclusively with robust features will offer superior estimates of breast cancer risk.

One potential criticism is that the AUCs observed in this study are lower than those in other published works [5,25]. However, the mean cross-validated AUC of 0.59 obtained here from the most robust class of features was not much lower than those in the work of Gastounioti et al. on the same dataset (cross-validated AUCs 0.59 to 0.67) [25]. Crucially, the risk-assessment models in Gastounioti’s work, and elsewhere in the literature, incorporate established demographic and clinical risk factors for breast cancer. We project that the best modeling practices described elsewhere, combined with a preference for low-CMV radiomic features, will further improve predictive performance.

One limitation of our study is the nature of the case-control analysis. In comparing the performance of features from one CMV quartile with those from another, we are necessarily comparing different features; moreover, different radiomic features will have different associations with cancer incidence regardless of CMV. Some radiomic features and combinations of features capture more information reflective of cancer risk, some less. We mitigate this weakness through, first, measuring AUC over multiple subsets of features within each quartile—reducing the impact of any single feature or feature combination with particularly good or bad predictive power; and second, via the mixed-model regression, assessing both the between-quartile and within-quartile variation in performance. However, the underlying limitation remains inherent to the nature of the comparison.

A further limitation is that, as the phantom was designed to simulate a CC view, we considered only CC views of clinical data in calculating the robustness metrics (Equations (1)–(3)). However, the data set used in the case-control analysis consisted predominately of MLO views, and we do not yet know how IAV and IWV are affected by CC versus MLO image data [25]. Further work could explore the effect of imaging view on feature robustness metrics.

An additional limitation of this paper is that we did not specifically identify the features that show so little variability in the population that they offer no predictive value in breast cancer risk assessment. Future work should explore strategies to identify such features; this strategy could potentially complement the robust feature selection method proposed in this paper as a way to improve cancer risk assessment.

A very important future extension will be to replicate the robustness calculations described in this study with data obtained with digital breast tomosynthesis (DBT) or “3D mammography”, a modality now increasingly replacing DM for breast cancer screening [29,30,31,32]. Future experiments with DBT would require a phantom suited for 3D imaging, as the Gammex-169 phantom was designed specifically for 2D x-ray imaging in the 1980s, well before the clinical implementation of DBT [16].

More generally, to ensure the broad applicability of the calculations described here, the work should be extended to more image types. For the many medical centers that do not retain copies of the raw (“FOR PROCESSING”) DM images, these should include processed (“FOR PRESENTATION”) DM images; other important types are individual projection images in a tomosynthesis acquisition (both raw and processed), a DBT reconstruction or reconstructed slice, and synthetic 2D images derived from a DBT scan [33]. Future work should also replicate the calculations described in this study with different software packages for texture feature extraction, for example, with an open-source software such as PyRadiomics or the Cancer Imaging Phenotypes Toolkit (CaPTk) [34,35,36]. It will also be important to validate the conclusions of this study on images from multiple facilities and multiple imaging device manufacturers.

## 5. Conclusions

The number of radiomic features being developed for risk-assessment modeling is constantly growing. However, having more features is not necessarily beneficial if the features are highly sensitive to imaging physics effects (contrast, noise, and image sharpness) and to non-meaningful feature differences, such as those between a woman’s left and right breasts. This paper proposes a method to identify robust radiomic features in DM by combining clinical data with phantom data acquired over a range of imaging acquisition settings. As new features are developed, this offers a method for selecting the features most likely to offer improvements in risk modeling—those which are more robust based on our measure of CMV.

## Figures and Tables

**Figure 1 cancers-13-05497-f001:**
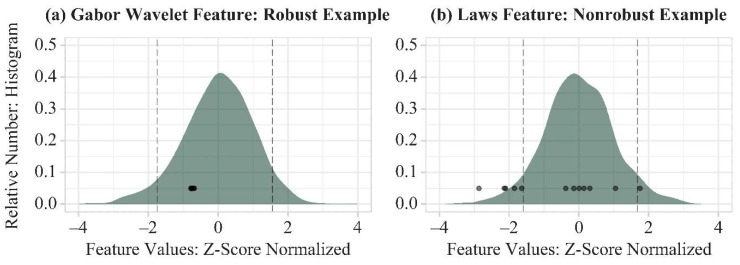
Histograms of the clinical distributions of two radiomic features. Superimposed points show the values obtained from imaging the phantom, with each data point corresponding to an x-ray acquisition (six unique kV/mAs combinations, repeated twice). (**a**) An example of a robust feature. The 12 phantom data points show very little variation relative to the clinical distribution; (**b**) An example of a nonrobust feature. The data derived from the phantom span a broader range.

**Figure 2 cancers-13-05497-f002:**
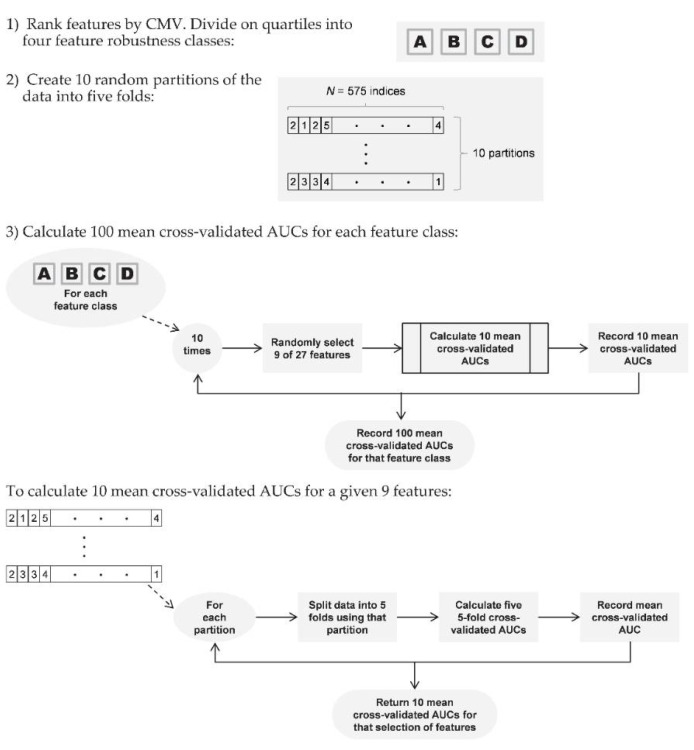
Flow diagram of AUC calculation on case-control data using logistic regression modeling.

**Figure 3 cancers-13-05497-f003:**
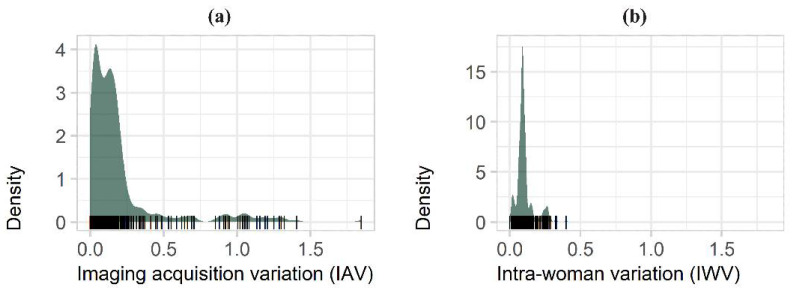
Summary of IAV and IWV results for all 341 features. (**a**) Histogram of IAV scores, with the long tail of features that are highly sensitive to imaging physics; (**b**) Histogram of IWV scores (this shows no such long tail).

**Figure 4 cancers-13-05497-f004:**
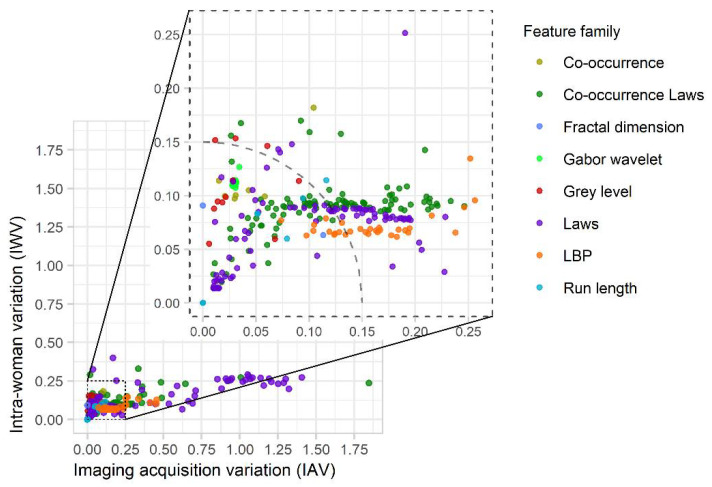
Summary of results obtained in IAV × IWV space for 341 features. The circular arc illustrates a threshold (CMV = 0.15) used to identify robust features (158 of 341 features, or 46%, are within this cutoff).

**Figure 5 cancers-13-05497-f005:**
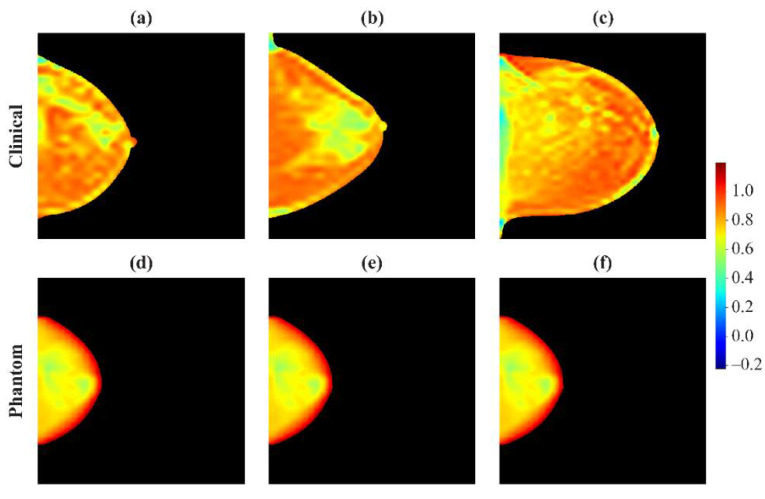
Heat maps for a robust feature (a Gabor Wavelet feature, CMV = 0.11) used as an example in Figure 1a. Top row: Clinical images at the (**a**) 5th, (**b**) 50th, and (**c**) 95th percentiles of the distribution in Figure 1a. Bottom row: Phantom images derived from various acquisition settings, corresponding to (**d**) left-most, (**e**) intermediate, and (**f**) right-most data points in Figure 1a. While there are clear differences between the three patients, the heat maps for the phantom are effectively identical. For the purposes of illustration, these figures are smoothed to avoid artifacts of pixelation.

**Figure 6 cancers-13-05497-f006:**
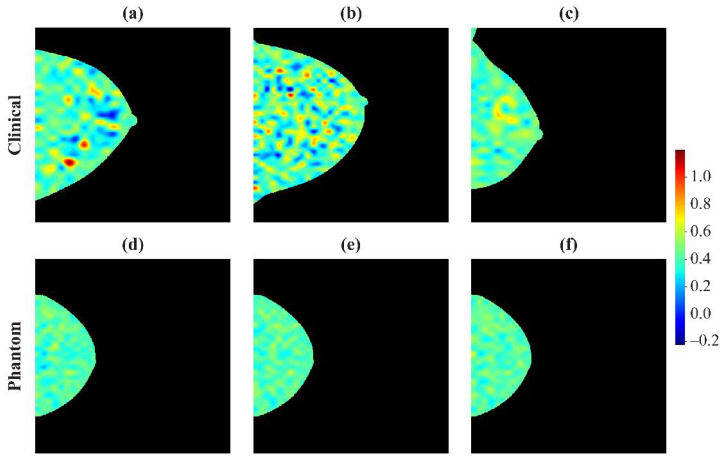
Heat maps for a non-robust feature (a Laws feature, CMV = 1.43) used as an example in Figure 1b. Top row: Heat maps derived from patients at the (**a**) 5th, (**b**) 50th, and (**c**) 95th percentiles of the distribution in Figure 1b. Bottom row: Heat maps illustrating how this feature is affected by variations in the imaging acquisition parameters, i.e., the (**d**) left-most, (**e**) intermediate, and (**f**) right-most data points in Figure 1b. Similar to Figure 5, these heat maps are smoothed to avoid artifacts of pixelation.

**Figure 7 cancers-13-05497-f007:**
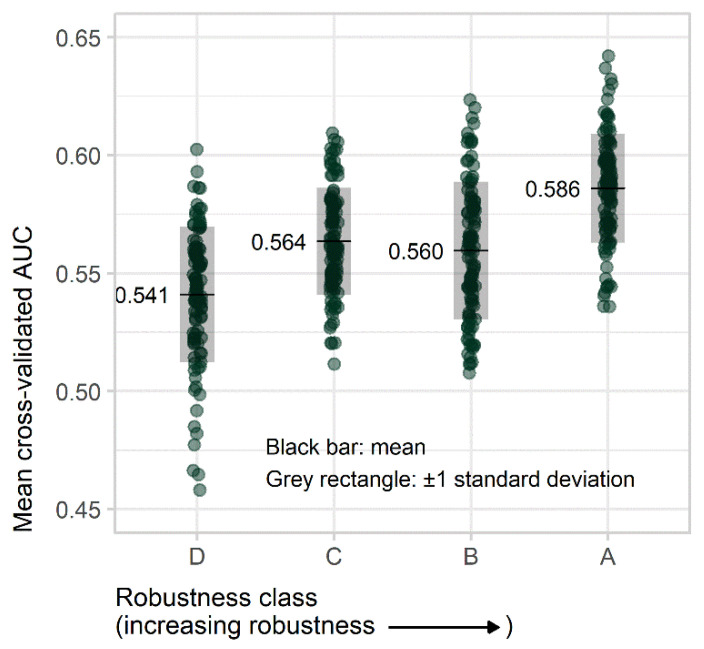
Mean five-fold cross-validated AUC results obtained with resampling techniques. There is an overall trend of higher AUC going from class D (least robust features) to A (most robust features). This demonstrates that more robust features offer superior case-control classification performance in risk-assessment calculations.

**Table 1 cancers-13-05497-t001:** Characteristics of participants used to calculate radiomic feature robustness metrics; both the source population and the subpopulation used for calculation of robustness metrics, with compressed breast thickness in [40, 60] mm.

Characteristic	Source Population: 997 Women *	Neither Breast with Thickness in [40, 60] mm: 445 Women	At Least One Breast with Thickness in [40, 60] mm: 552 Women ^†^	*p*-Value ^‡^
**Age**				0.01
<40	29 (2.9%)	18 (4.0%)	11 (2.0%)	
[40, 50)	270 (27.1%)	128 (28.8%)	142 (25.7%)	
[50, 60)	304 (30.5%)	144 (32.4%)	160 (29.0%)	
[60, 70)	268 (26.9%)	113 (25.4%)	155 (28.1%)	
≥70	126 (12.6%)	42 (9.4%)	84 (15.2%)	
**Race**				0.96
Black	464 (46.5%)	206 (46.3%)	258 (46.7%)	
White	457 (45.8%)	207 (46.5%)	250 (45.3%)	
Other	64 (6.4%)	28 (6.3%)	36 (6.5%)	
Missing	12 (1.2%)	4 (0.9%)	8 (1.4%)	
**BI-RADS** ^ **®** ^ **Density**				<0.005
A	113 (11.3%)	58 (13.0%)	55 (10.0%)	
B	552 (55.4%)	258 (58.0%)	294 (53.3%)	
C	310 (31.1%)	113 (25.4%)	197 (35.7%)	
D	22 (2.2%)	16 (3.6%)	6 (1.1%)	
**BMI**, Median (IQR)	28.1	26.7	29.9	<0.005
	(23.8–33.6)	(23.5–32.3)	(24.6–35.3)	

* Population used for radiomic feature dimensionality reduction steps described in Section 2.3.2. ^†^ Subpopulation used for calculation of radiomic feature robustness metrics: IAV, IWV, and CMV. ^‡^ *p*-Value by Fisher’s exact test for categorical covariates, Mann–Whitney-Wilcoxon test for BMI.

**Table 2 cancers-13-05497-t002:** Characteristics of participants in independent data set used to test the performance of radiomic features in case-control regression.

Characteristic	Total Population: 575 Women	Cases: 115 Women	Controls: 460 Women	*p*-Value *
**Age**				1
<40	10 (1.7%)	2 (1.7%)	8 (1.7%)	
[40, 50)	145 (25.2%)	29 (25.2%)	116 (25.2%)	
[50, 60)	135 (23.5%)	27 (23.5%)	108 (23.5%)	
[60, 70)	180 (31.3%)	36 (31.3%)	144 (31.3%)	
≥70	105 (18.3%)	21 (18.3%)	84 (18.3%)	
**Race**				1
Black	305 (53.0%)	61 (53.0%)	244 (53.0%)	
White	270 (47.0%)	54 (47.0%)	216 (47.0%)	
**BI-RADS** ^ **®** ^ **Density**				0.08
A	63 (11.0%)	9 (7.8%)	54 (11.7%)	
B	340 (59.1%)	61 (53.0%)	279 (60.7%)	
C	161 (28.0%)	38 (33.0%)	123 (26.7%)	
D	6 (1.0%)	3 (2.6%)	3 (0.7%)	
Missing	5 (0.9%)	4 (3.5%)	1 (0.2%)	
**BMI**, Median	28.3	28.1	29	0.54
(IQR)	(23.7–34.6)	(23.6–34.7)	(24.2–34.5)	

* *p*-value by Fisher’s exact test for categorical covariates, Mann–Whitney–Wilcoxon test for BMI.

**Table 3 cancers-13-05497-t003:** Statistical analysis of AUC results in four robustness classes using mixed model logistic regression. The reference class (D) consists of the least robust features. Classes with more robust features (A, B, and C) tend to offer statistically-higher AUCs in case-control classification.

Robustness Class	CMV Range (Mean)	AUC Mean (SD)	Coefficient (95% CI)	Standard Error	*p*-Value *
**D**	0.47–1.43 (0.97)	0.54 (0.029)	*Reference*		
**C**	0.19–0.46 (0.30)	0.56 (0.023)	0.022 (0.015–0.030)	0.004	<0.005
**B**	0.14–0.19 (0.16)	0.56 (0.029)	0.018 (0.011–0.026)	0.004	<0.005
**A**	0.020–0.14 (0.089)	0.59 (0.023)	0.045 (0.038–0.052)	0.004	<0.005

* Versus null hypothesis that coefficient is zero. SD of random effects: 0.004. Model *p*-value < 0.005 (versus null hypothesis that AUC is unrelated to robustness class).

## Data Availability

The data generated during the current study are available from the corresponding author on reasonable request.

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
