# Peer review of "Incorporating Robustness to Imaging Physics into Radiomic Feature Selection for Breast Cancer Risk Estimation"

_cancers, 2021, doi:10.3390/cancers13215497_

Round 1

Reviewer 1 Report

In the manuscript #cancers-1328303, the authors implement radiomics approach to investigate the breast cancer risk estimation using feature selection. The aim of this study was to predict of breast cancer risk using radiomics texture features of mammographic density. Radiomics approaches are relevant to the aims and scopes of cancers and have high interest for the reading audience. 

Major:

  • Radiomics features were 341 image textures. Robustness was estimated 32 Gabor wavelet features. An analysis on the image texture feature group that can specifically evaluate the robustness from each intra-woman variation and inter-woman variation were not performed.
  • Logistic regression test accuracy should be estimated by robust and nonrobust feature group.
  • The reason for using Robustness Class D as a reference is not described, please describe it.

Minor:

  • The number of validation and test data set should be described.
  • In line 165, 12 data pointes were mentioned. It should be described how set the data point.

Reviewer 2 Report

Title Clearly defines the study objective

Abstract The abstract is easy to read and allows even a less experienced reader to understand the methods and main findings of the study.

MAIN TEXT The description of the materials methods is very articulated, it could be useful to try to insert a summary schema/paragraph to allow the reader not to get lost in the vast amount of details provided.

The section of the results is also rich but it is not well underlined which are the results that the authors consider more important for each type of feature studied. It is advisable to add an outline.

DISCUSSION AND CONCLUSIONS The discussion is well conducted and it also includes the limitations of the study as well as future potential.

It would be interesting to give more space to the potential future clinical impact of the results of this study.

Otherwise, it would remain a purely descriptive work of a technical aspect without a potential future scenario.

Round 2

Reviewer 2 Report

I suggest to provide minor spell check